# Behavioral Cloning from Noisy Demonstrations

**Fumihiro Sasaki & Ryota Yamashina**
Ricoh Company, Ltd.
{fumihiro.fs.sasaki,ryohta.yamashina}@jp.ricoh.com

## Abstract

We consider the problem of learning an optimal expert behavior policy given noisy demonstrations that contain observations from both optimal and non-optimal expert behaviors. Popular imitation learning algorithms, such as generative adversarial imitation learning, assume that (clean) demonstrations are given from optimal expert policies but not the non-optimal ones, and thus often fail to imitate the optimal expert behaviors given the noisy demonstrations. Prior works that address the problem require (1) learning policies through environment interactions in the same fashion as reinforcement learning, and (2) annotating each demonstration with confidence scores or rankings. However, such environment interactions and annotations in real-world settings take impractically long training time and a significant human effort. In this paper, we propose an imitation learning algorithm to address the problem without any environment interactions and annotations associated with the non-optimal demonstrations. The proposed algorithm learns ensemble policies with a generalized behavioral cloning (BC) objective function where we exploit another policy already learned by BC. Experimental results show that the proposed algorithm can learn behavior policies that are much closer to the optimal policies than ones learned by BC.

## 1 Introduction

Imitation learning (IL) has become a widely used approach to obtain autonomous robotics control systems. IL is often more applicable in real-world problems than reinforcement learning (RL) since expert demonstrations are often easier than designing appropriate rewards that RL requires. There have been several IL methods that involve RL (Ziebart et al., 2008; Ng et al., 2000; Abbeel & Ng, 2004; Ho & Ermon, 2016). Those IL methods inherit sample complexity from RL in terms of environment interactions during training. The complexity restricts applicabilities in real-world problems since a number of environment interactions in real-world settings often take a long time and cause damage to the robot or the environment. Therefore, we are interested in IL methods that do not require the environment interactions, such as behavioral cloning (BC) (Pomerleau, 1991) which learns an expert policy in a supervised fashion.

BC as well as popular IL methods, such as generative adversarial imitation learning (GAIL) (Ho & Ermon, 2016), assume the expert demonstration is optimal. Unfortunately, it is often difficult to obtain optimal demonstrations for many tasks in real-world problems because the expert who tries to operate the robot so that it can achieve tasks often makes mistakes due to various reasons, such as the difficulty of the task, difficulty in handling the controller, limited observability of the environment, or the presence of distraction. The mistakes include unnecessary and/or incorrect operations to achieve the tasks. Given such *noisy* expert demonstrations, which contain records of both optimal and non-optimal behavior, BC as well as the popular IL methods fails to imitate the optimal policy due to the optimal assumption on the demonstrations as shown in (Wu et al., 2019).

A naive solution to cope with the noisy demonstrations is discarding the non-optimal demonstrations among the ones that were already collected. This screening process is often impractical because it involves a significant human effort. Most of recent IL works suppose settings where a very limited number of *clean* expert demonstrations, which are composed of only the optimal behavior records, are available. Those methods are also vulnerable to the noisy demonstrations due to the optimal

assumption on the demonstrations. Thus they implicitly suppose such impractical screening process if they were applied in real-world problems, where a number of the noisy demonstrations other than the clean ones can be easily obtained. There have been IL methods addressing the noisy demonstrations. Instead of the screening process, they require to annotate each demonstration with confidence scores (Wu et al., 2019) or rankings (Brown et al., 2019). Even though they cope well with the noisy demonstrations to obtain the optimal behavior policies, such annotation costs a significant human effort as it is for the screening. Hence, we desire IL methods that can cope well with the noisy demonstrations, which can be easily obtained in real-world settings, without any screening and annotation processes associated with the non-optimal behaviors.

In this paper, we propose a novel imitation learning algorithm to address the noisy demonstrations. The proposed algorithm does not require (1) any environment interactions during training, and (2) any screening and annotation processes associated with the non-optimality of the expert behaviors. Our algorithm learns ensemble policies with a generalized BC objective function where we exploit another policy already learned by BC. Experimental results show that the proposed algorithm can learn policies that are much closer to the optimal than ones learned by BC.

## 2 RELATED WORKS

A wide variety of IL methods have been proposed in these last few decades. BC (Pomerleau, 1991) is the simplest IL method among those and thus BC could be the first IL option when enough clean demonstrations are available. Ross & Bagnell (2010) have theoretically pointed out a downside of the BC which is referred to as *compounding error* – the small errors of the learners trained by BC could compound over time and bring about the deterioration of their performance. On the other hand, experimental results in (Sasaki et al., 2018) show that BC given the clean demonstrations of sufficient amounts can easily obtain the optimal behavior even for complex continuous control tasks. Hence, the effect of the compounding error is negligible in practice if the amount of clean demonstrations is sufficient. However, even if the amount of the demonstrations is large, BC cannot obtain the optimal policy given the noisy demonstrations due to the optimal assumption on the demonstrations. Another widely used IL approaches are inverse reinforcement learning (IRL) (Ziebart et al., 2008; Ng et al., 2000; Abbeel & Ng, 2004) and adversarial imitation learning (AIL) (Ho & Ermon, 2016). Since those approaches also assume the optimality of the demonstrations, they are also not able to obtain the optimal policy given the noisy demonstrations, as shown in (Wu et al., 2019). As we will show in Section 6, our algorithm successfully can learn near-optimal policies if noisy demonstrations of sufficient amounts are given.

There have been several works that address the noisy demonstrations (Wu et al., 2019; Brown et al., 2019; Tangkaratt et al., 2019; Kaiser et al., 1995; Grollman & Billard, 2012; Kim et al., 2013). Those works address the noisy demonstrations by either screening the non-optimal demonstrations with heuristic non-optimal assessments (Kaiser et al., 1995), annotations associated with the non-optimality (Wu et al., 2019; Brown et al., 2019; Grollman & Billard, 2012), or training through the environment interactions (Kim et al., 2013; Wu et al., 2019; Brown et al., 2019; Tangkaratt et al., 2019). Our algorithm does not require any screening processes, annotations associated with the non-optimality, and the environment interactions during training.

Offline RL methods (Lange et al., 2012; Fujimoto et al., 2019; Kumar et al., 2020) train the learner agents without any environment interactions, and allow the training dataset to have non-optimal trajectories as in our problem setting. A drawback of offline RL methods for the real-world applications is the requirement to design reward functions, which often involves a significant human efforts for its success, since those methods assume that the reward for each state-action pair is known. Our algorithm does not require to design reward functions as in standard IL methods.

Disagreement regularized imitation learning (DRIL) (Brantley et al., 2019) is a state-of-the-art IL algorithm which employs an ensemble of policies as our algorithm does. The aims of employing the ensemble is different between DRIL and our algorithm. DRIL uses the disagreement in predictions made by policies in the ensemble to evaluate whether the *states* observed during training the learner are ones observed in the expert demonstrations. On the other hand, our algorithm uses the ensemble to encourage the learner to take optimal *actions* on each state as described in 5.3. In addition, DRIL fundamentally requires the environment interactions during training whereas our algorithm does not.

## 3 PRELIMINARIES AND PROBLEM SETUP

In this work, we consider an episodic fixed-horizon Markov decision process (MDP) which is formalized as a tuple $\{\mathcal{S}, \mathcal{A}, \mathcal{P}, R, d_0, T\}$, where $\mathcal{S}$ is a set of states, $\mathcal{A}$ is a set of possible actions agents can take, $\mathcal{P} : \mathcal{S} \times \mathcal{A} \times \mathcal{S} \to [0, 1]$ is a transition probability, $R : \mathcal{S} \times \mathcal{A} \to [0, 1]$ is a reward function, $d_0 : \mathcal{S} \to [0, 1]$ is a distribution over initial states, and $T$ is an episode horizon. The agent's behavior is defined by a stochastic policy $\pi : \mathcal{S} \times \mathcal{A} \to [0, 1]$ and $\Pi$ denotes a set of the stochastic policies. The expected one-step immediate reward for a policy $\pi$ given a state $s$ is defined as $R^\pi(s) = \mathbb{E}_{a \sim \pi(\cdot|s)}[R(s, a)]$.

Let $d_t^\pi$ and $d^\pi = \frac{1}{T} \sum_{t=1}^{T} d_t^\pi$ denote the distribution over states at time step $t$ and the average distribution over $T$ time steps induced by $\pi$, respectively. The distributions $d_1^\pi$ at the first step correspond to $d_0$ for any $\pi$. When following a policy $\pi$ throughout an episode, the expected one-step immediate reward at time step $t$ and the expected $T$-step reward are defined as $R_t^\pi = \mathbb{E}_{s \sim d_t^\pi, a \sim \pi(\cdot|s)}[R(s, a)] = \mathbb{E}_{s \sim d_t^\pi}[R^\pi(s)]$ and $\mathcal{J}(\pi, R) = \sum_{t=1}^{T} R_t^\pi = T \mathbb{E}_{s \sim d^\pi}[R^\pi(s)]$, respectively. We refer to $\mathcal{J}(\pi, R)$ as *on-policy* expected $T$-step reward. We also consider another $T$-step reward defined by $\mathcal{J}_\beta(\pi, R) = T \mathbb{E}_{s \sim d^\beta}[R^\pi(s)]$, which we call *off-policy* expected $T$-step reward, where $\beta \in \Pi$ is a policy that can differ from $\pi$.

In our problem setting, the functions $R$ is not given. Instead, we observe noisy demonstrations. We refer to the agent that generates the noisy demonstrations as the *noisy expert*. The decision process turns to be MDP$\setminus\{R\}$ as in the common imitation learning settings, and our problem can be formalized as to find an optimal policy in MDP$\setminus\{R\}$. Here we refer to the *true expert* policy $\pi_e^*$ as ones being able to take the optimal (thus not noisy) behavior in episodic tasks. We make the following four assumptions to further formalize our problem setting:

**Assumption 1.** *The $T$-step expected reward of $\pi_e^*$ satisfies $\mathcal{J}(\pi, R) \leq \mathcal{J}(\pi_e^*, R)$; $\mathcal{J}_\beta(\pi, R) \leq \mathcal{J}_\beta(\pi_e^*, R)$; and $\mathcal{J}_\beta(\pi_e^*, R) \leq \mathcal{J}(\pi_e^*, R)$ for any non-optimal policies $\pi, \beta \in \Pi \setminus \{\pi_e^*\}$.*

**Assumption 2.** *With small probability $\epsilon$, which we call non-optimal probability, the policies $\pi_e$ the noisy experts follow during demonstrations are sampled at each time step as $\pi_e = \pi \sim p_\Pi$ if $\epsilon \geq z \sim \mathcal{U}(0, 1)$, otherwise $\pi_e = \pi_e^*$, where $p_\Pi$ is an unknown distribution over the set of policies, $z$ is a random variable, and $\mathcal{U}(0, 1)$ is a uniform distribution with range $[0, 1]$.*

**Assumption 3.** *The reward $R_t^{\pi_e}$ is at least zero if the noisy expert has followed a policy $\pi \in \Pi \setminus \{\pi_e^*\}$ once or more so far, otherwise $R_t^{\pi_e} = \mathbb{E}_{s \sim d_t^{\pi_e}}[\epsilon \mathbb{E}_{\pi \sim p_\Pi}[R^\pi(s)] + (1 - \epsilon)R^{\pi_e^*}(s)]$.*

**Assumption 4.** *The sequence $\{R_1^{\pi_e}, ..., R_T^{\pi_e}\}$ has monotonically decreasing property $R_t^{\pi_e} \geq R_{t+1}^{\pi_e}$.*

Assumption 1 indicates that both on-policy and off-policy expected $T$-step reward following $\pi_e^*$ are always greater than or equal to ones following any other policies. In other words, we assume the true expert policy is an optimal one in the MDP, and the agent following the policy is able to behave so that the expected immediate rewards at any states are maximized. Under Assumption 1, the problem that we would like to solve in this work can be said to learn a parameterized policy $\pi_\theta$ to maximize its *on*-policy expected $T$-step reward $\mathcal{J}(\pi_\theta, R)$ to $\mathcal{J}(\pi_e^*, R)$. Assumption 2 indicates that the noisy expert occasionally adopts non-optimal policies, which results in the noisy demonstrations, due to random events, such as the presence of distractions, associated with the random variable $z$. The noisy expert is going to visit states that would be never visited by the true expert if the noisy expert followed non-optimal policies even once. Assumption 3 indicates that those states are less rewarded and their rewards are at least zero. Assumption 3 also indicates that the noisy demonstrations have a number of episodes where the noisy expert has reached the same state $s$ where the noisy expert has adopted both $\pi_e^*$ and $\pi \in \Pi \setminus \{\pi_e^*\}$ with the probability $\epsilon$. Assumption 4 indicates that, since the probability the noisy expert consecutively follows $\pi_e^*$ decreases as time step increases according to Assumption 2, the divergence between $d_t^{\pi_e}$ and $d_t^{\pi_e^*}$ becomes greater as the number of time step $t$ increases, and thus the one-step expected immediate reward $R_t^{\pi_e} = \mathbb{E}_{s \sim d_t^{\pi_e}, a \sim \pi_e(\cdot|s)}[R(s, a)]$ decreases as $t$ increases.

# 4 ANALYSIS OF PERFORMANCE DETERIORATION

In this section, we firstly describe BC objective in 4.1. Then, we analyze why the learner trained by BC deteriorates its performance when using the noisy demonstrations from *the expected T-step reward maximization* and *KL-divergence minimization* perspectives in 4.2 and 4.3, respectively.

## 4.1 BEHAVIORAL CLONING OBJECTIVE

Let $\pi_\theta \in \Pi$ is a learner policy parameterized by $\theta$ to be optimized by IL algorithms. The objective of BC in common is as follows:

$$\arg\max_\theta \mathbb{E}_{s \sim d^{\pi_e}, a \sim \pi_e(\cdot|s)}[\log \pi_\theta(a|s)]. \tag{1}$$

The objective (1) aims to mimic the expert behavior which follows $\pi_e$. It can be interpreted that (1) is to maximize the expected one-step immediate reward $R^{\pi_\theta}(s)$ to $R^{\pi_e}(s)$ at each state $s \sim d^{\pi_e}$. Since the state distribution $d^{\pi_e}$ is not induced by $\pi_\theta$, it can also be said that (1) is to maximize the off-policy expected $T$-step rewards $\mathcal{J}_{\pi_e}(\pi_\theta, R)$ to $\mathcal{J}(\pi_e, R)$.

## 4.2 THE EXPECTED T-STEP REWARD MAXIMIZATION

We obtain the lower bound of the expected on-policy $T$-step reward for the noisy expert policy in almost the same way to derive Theorem 2.1 in (Ross & Bagnell, 2010) where they showed the lower bound for the learner policies given the "clean" expert demonstrations.

**Theorem 1.** *If the Assumptions 1 - 4 hold, $\mathcal{J}(\pi_e, R)$ has the following lower bound:*

$$\mathcal{J}(\pi_e, R) \geq \left\{ \frac{1}{T} \sum_{t=0}^{T-1} (1 - \epsilon)^t \right\} \cdot \mathbb{E}_{\pi \sim p_\Pi}[\mathcal{J}_{\pi_e}(\pi, R)]. \tag{2}$$

The detailed derivation can be found in Appendix A.1. Assume that the learner policy $\pi_\theta$ has a probability of non-optimal behavior $\hat{\epsilon} = \epsilon + \zeta$ at most as the result of BC, where $\zeta \in [0, 1 - \epsilon]$ is an additional probability of non-optimal behavior due to the remained loss in (1). Note that $\zeta$ may become greater than zero due to the difficulty in the optimization of (1) even if $\epsilon = 0$. The learner following $\pi_\theta$ with $\hat{\epsilon}$ can be deemed as another noisy expert who samples a policy at each time step $\pi_\theta = \pi \sim p_{\pi_\theta}$ if $\hat{\epsilon} \geq z \sim \mathcal{U}(0, 1)$, otherwise $\pi_\theta = \pi_e^*$, where $p_{\pi_\theta}$ is a (special) distribution from which the same policy is always sampled. By replacing $\hat{\epsilon}$ and $p_{\pi_\theta}$ from $\epsilon$ and $p_\Pi$ in Theorem 1 respectively, we obtain the following corollary.

**Corollary 1.** *If the Assumptions 1 - 4 hold and the policy $\pi_\theta$ has a probability of non-optimal behavior $\hat{\epsilon} = \epsilon + \zeta$, $\mathcal{J}(\pi_\theta, R)$ has the following lower bound:*

$$\mathcal{J}(\pi_\theta, R) \geq \left\{ \frac{1}{T} \sum_{t=0}^{T-1} (1 - \hat{\epsilon})^t \right\} \cdot \mathcal{J}_{\pi_e}(\pi_\theta, R). \tag{3}$$

Recall that the BC objective (1) is to maximize $\mathcal{J}_{\pi_e}(\pi_\theta, R)$. If $\hat{\epsilon} = 0$, Corollary 1 indicates that the *on*-policy expected $T$-step reward $\mathcal{J}(\pi_\theta, R)$, which corresponds to the actual learner performance, is boosted by maximizing $\mathcal{J}_{\pi_e}(\pi_\theta, R)$ through the optimization of the BC objective (1). On the other hand, if $\epsilon > 0$ and thus $\hat{\epsilon} > 0$, the first factor on the RHS in (3) becomes much smaller as $\epsilon$ becomes larger. Corollary 1 thus shows that the probability of non-optimal behavior $\epsilon$ of the noisy expert significantly negates the improvement of learner performance $\mathcal{J}(\pi_\theta, R)$ by BC even if $\zeta$ can be sufficiently minimized through the optimization. Hence, the learner trained by BC is not able to boost the learner performance enough if the noisy demonstrations were given.

## 4.3 KL DIVERGENCE MINIMIZATION

Let $S^{\pi_e}$ be a set of states that are observed in the noisy demonstration. $S^{\pi_e}$ can be thought of as the domain of (empirical) state distributions $d^{\pi_e}$. $S^{\pi_e}$ can be defined with two state sets of states as $S^{\pi_e} = S_e^{\pi_e} \cup S_{e+*}^{\pi_e}$, where $S_e^{\pi_e}$ contains states that are observed if the noisy expert has followed a policy $\pi \in \Pi \setminus \{\pi_e^*\}$ once or more so far in the episode, and $S_{e+*}^{\pi_e}$ contains states

at which the noisy expert has followed a policy $\pi \in \Pi \setminus \{\pi_e^*\}$ at the first time in the episode. Under Assumption 3, the rewards $R_t^{\pi_e}$ for the states $s \in S_e^{\pi_e}$ are at least zero whereas $R_t^{\pi_e} = \mathbb{E}_{s \sim d_t^{\pi_e}}\left[\epsilon \mathbb{E}_{\pi \sim p_\Pi}[R^\pi(s)] + (1 - \epsilon) R^{\pi_e^*}(s)\right]$ for the states $s \in S_{e+*}^{\pi_e}$. Note that the noisy expert adopts $\pi \in \Pi \setminus \{\pi_e^*\}$ with a probability $\epsilon$ at the states $s \in S_{e+*}^{\pi_e}$. Let $d_e^{\pi_e}$ and $d_{e+*}^{\pi_e}$ be the state distributions the noisy expert policy induces in $S_e^{\pi_e}$ and $S_{e+*}^{\pi_e}$, respectively. Then we can define $d^{\pi_e}$ as a mixture of those distributions as

$$d^{\pi_e}(s) = \alpha d_e^{\pi_e}(s) + \beta d_{e+*}^{\pi_e}(s), \tag{4}$$

where $\alpha$ and $\beta$ are ratios the noisy expert entered states that belong to $S_e^{\pi_e}$ and $S_{e+*}^{\pi_e}$ during demonstrations, respectively. In addition, $\alpha + \beta = 1$ is satisfied. Using Equation (4), the upper bound of the objective function in Equation (1) is derived as follows:

$$\mathbb{E}_{s \sim d^{\pi_e}, a \sim \pi_e(\cdot|s)}[\log \pi_\theta(a|s)] \leq -\alpha \Omega_e(\theta) - \beta \Omega_{e+*}(\theta), \tag{5}$$

$$\Omega_e(\theta) = \mathbb{E}_{s \sim d_e^{\pi_e}}[D_{KL}[\pi_e(\cdot|s)||\pi_\theta(\cdot|s)]], \tag{6}$$

$$\Omega_{e+*}(\theta) = \epsilon \mathbb{E}_{s \sim d_{e+*}^{\pi_e}, \pi \sim p_\Pi}[D_{KL}[\pi(\cdot|s)||\pi_\theta(\cdot|s)]]$$
$$+ (1 - \epsilon) \mathbb{E}_{s \sim d_{e+*}^{\pi_e}}[D_{KL}[\pi_e^*(\cdot|s)||\pi_\theta(\cdot|s)]], \tag{7}$$

where $D_{KL}$ is forward Kullback-Leibler (KL) divergence. The full derivation can be found in Appendix A.2. The inequality (5) shows that the BC objective (1) with the noisy demonstrations is to minimizes the sum of KL divergences. The first term on the RHS in (7) leads the learner to imitate some non-optimal behaviors whereas the second term is to learn $\pi_e^*$ on the same states. The optimization to maximize the RHS in (7) is difficult because minimizing KL divergences with different target distributions at the same time is difficult in general. The first term on the RHS in (7) thus works as a *"noisy" regularizer* with a coefficient $\epsilon$ that makes the learner confused to learn $\pi_e^*$. The difficulty in the optimization due to the noisy regularizer increases $\zeta$ as $\epsilon$ increases.

As mentioned in 4.1 and 4.2, BC is to maximize $\mathcal{J}_{\pi_e}(\pi_\theta, R)$ to $\mathcal{J}(\pi_e, R)$. Hence, minimizing $\Omega_e(\theta)$ in (6) corresponds to maximize $\mathbb{E}_{s \sim d_e^{\pi_e}}[R^{\pi_\theta}(s)]$ to $\mathbb{E}_{s \sim d_e^{\pi_e}}[R^{\pi_e}(s)]$. Since the rewards $R^{\pi_e}(s)$ are at least zero for the states $s \sim d_e^{\pi_e}$ according to Assumption 3 and the definition of $S_e^{\pi_e}$, $\mathbb{E}_{s \sim d_e^{\pi_e}}[R^{\pi_\theta}(s)]$ becomes at least zero by minimizing $\Omega_e(\theta)$. Hence $\mathcal{J}_{\pi_e}(\pi_\theta, R)$ becomes at least zero as the rate $\alpha$ increases, while the rate $\alpha$ increases as the probabilities of non-optimal behavior $\epsilon$ increases. Thus, the larger the probability $\epsilon$ is, the more difficult it is to boost the learner performance by BC.

If the influence of the noisy regularizer can be reduced, probabilities the learner follows $\pi_e^*$ at the state $s \in S_{e+*}^{\pi_e}$ will increase. In addition, as probabilities the learner follows $\pi_e^*$ at the states $s \in S_{e+*}^{\pi_e}$ increase, the rate (corresponding to $\alpha$) for the states $s \in S_e^{\pi_e}$ will decrease. Thus, it can be said that, the more often learner follows $\pi_e^*$ at the states $s \in S_{e+*}^{\pi_e}$, the more rewards $R^{\pi_e}(s)$ the learner obtains according to Assumption 3. To summarize the above analysis, reducing the influence of the noisy regularizer for states $s \in S_{e+*}^{\pi_e}$, which leads the learner to imitate some non-optimal behaviors, might boost the learner performance.

# 5 ALGORITHM

The analyses in Section 4 describe that the learner trained by standard BC deteriorates its performance when the noisy demonstrations are given. Based on both analyses in 4.2 and 4.3, the learner performance will be boosted if the learner imitates the optimal policy $\pi_e^*$ but not the non-optimal ones $\pi \in \Pi \setminus \{\pi_e^*\}$ for the states $s \in S_{e+*}^{\pi_e}$. In other words, the learner performance will be boosted if $\hat{\epsilon}$ of the learner can be reduced. In this section, we first propose our algorithm that avoids learning $\pi \in \Pi \setminus \{\pi_e^*\}$ while learning $\pi_e^*$ in 5.1. Then we describe how our algorithm works to avoid learning $\pi \in \Pi \setminus \{\pi_e^*\}$ from *mode seeking* and *reward maximization* perspectives in 5.2 and 5.3, respectively. We lastly provide limitations of our algorithm in 5.4.

## 5.1 PROPOSED ALGORITHM

We consider a generalization of the BC objective as follows:

$$\arg \max_\theta \mathbb{E}_{s \sim d^{\pi_e}, a \sim \pi_e(\cdot|s)}[\log \pi_\theta(a|s) \cdot \hat{R}(s, a)], \tag{8}$$

---

**Algorithm 1** Behavioral Cloning from Noisy Demonstrations

---

1: Given the expert demonstrations $\mathcal{D}$.
2: Set $\hat{R}(s, a) = 1$ for $\forall (s, a) \in \mathcal{D}$.
3: Split $\mathcal{D}$ into $K$ disjoint sets $\{\mathcal{D}^1, \mathcal{D}^2, ..., \mathcal{D}^K\}$.
4: **for** iteration = 1, $M$ **do**
5:    **for** $k = 1, K$ **do**
6:      Initialize parameters $\theta^k$.
7:      **for** $l = 1, L$ **do**
8:        Sample a random minibatch of $N$ state-action pairs $(s_n, a_n)$ from $\mathcal{D}^k$.
9:        Calculate a sampled gradient $\frac{1}{N} \sum_{n=1}^{N} \nabla_{\theta^k} \log \pi_{\theta^k}(s_n, a_n) \cdot \hat{R}(s_n, a_n)$.
10:       Update $\theta^k$ by gradient ascent using the sampled gradient.
11:      **end for**
12:    **end for**
13:    Copy $\pi_{\theta_{old}} \leftarrow \pi_\theta$.
14:    Set $\hat{R}(s, a) = \pi_{\theta_{old}}(a|s)$ for $\forall (s, a) \in \mathcal{D}$.
15: **end for**
16: **return** $\pi_\theta$.

---

where $\hat{R} : \mathcal{S} \times \mathcal{A} \to [0, 1]$ denotes an arbitrary function which can differ from $R$. If $\hat{R}(s, a) = 1$ for $\forall (s, a) \in \mathcal{S} \times \mathcal{A}$, the objective (8) corresponds to the BC objective (1). If $\int_\mathcal{A} \hat{R}(s, a)da = 1$ for $\forall s \in \mathcal{S}$ is satisfied, $\hat{R}(s, a)$ can be interpreted as weights for action samples obtained by the demonstrations so that the actions are sampled according to their relative weights. The objective (8) can also be deemed as that of the off-policy actor-critic (Off-PAC) algorithm[1] (Degris et al., 2012) with reward functions $\hat{R}(s, a)$ and zero discount factors.

Let $\pi_{\theta^1}, \pi_{\theta^2}, ..., \pi_{\theta^K}$ be $K$ parameterized policies with different initial parameters $\theta^1, \theta^2, ..., \theta^K$, and $\pi_\theta(a|s) = \sum_{k=1}^{K} \pi_{\theta^k}(a|s)/K$ denotes an ensemble of the parameterized policies with parameters $\theta = \{\theta^1, \theta^2, ..., \theta^K\}$. Let $\pi_{\theta_{old}}$ be a parameterized policy with $\theta_{old}$ which was already optimized with the noisy demonstrations. The main idea of our algorithm is to reuse the old policy $\pi_{\theta_{old}}$ as $\hat{R}(s, a)$ in the generalized BC objective (8).

$$\arg \max_\theta \mathbb{E}_{s \sim d^{\pi_e}, a \sim \pi_e(\cdot|s)}[\log \pi_\theta(a|s) \cdot \pi_{\theta_{old}}(a|s)]. \tag{9}$$

The overview of our algorithm is described in Algorithm 1.

## 5.2 WEIGHTED ACTION SAMPLING FOR $\pi_e^*$ MODE SEEKING

Since $\pi_{\theta_{old}}$ satisfies $\int_\mathcal{A} \pi_{\theta_{old}}(a|s)da = 1$ for $\forall s \in \mathcal{S}$, $\pi_{\theta_{old}}$ can be interpreted as the weights for the weighted action sampling. We below explain the weighted action sampling procedure in our algorithm on $S_{e+*}^{\pi_e}$. Figure 1 depicts a toy example of the sampling procedure. The distribution of the noisy expert actions on $S_{e+*}^{\pi_e}$ is a mixture of two distributions as shown in Equation (7). If $\epsilon$ is sufficiently small, $\pi_\theta$ is optimized so that its mode is closer to that of $\pi_e^*$ than $\pi \in \Pi \setminus \{\pi_e^*\}$ according to mode seeking properties of the forward KL divergence (Ghasemipour et al., 2020). Given the sampling weights $\pi_{\theta_{old}}(a|s) = \pi_\theta(a|s)$ for the empirical action samples, the weighted action distribution distorts so that its mode also gets closer to the mode of $\pi_e^*$. By iteratively distorting the weighted action distribution with the same procedure, its mode fits to near the mode of $\pi_{e*}$. The weights for actions sampled from $\pi \in \Pi \setminus \{\pi_e^*\}$ eventually become much smaller, and thus the learner will not learn $\pi \in \Pi \setminus \{\pi_e^*\}$. The mode seeking procedure of our algorithm is analogous to the mean shift algorithm (Fukunaga & Hostetler, 1975) so that the mode of $\pi_\theta$ shifts towards that of $\pi_e^*$ by minimizing the KL divergence between $\pi_\theta$ and the weighted action distribution.

---

[1]Although Off-PAC multiplies $\log \pi_\theta(a|s)$ by a density ratio $\pi_e(s|a)/\pi_\theta(s|a)$, $\pi_\theta(s|a)$ is empirically approximated to be one in popular off-policy RL algorithms such as DDPG (Lillicrap et al., 2015).

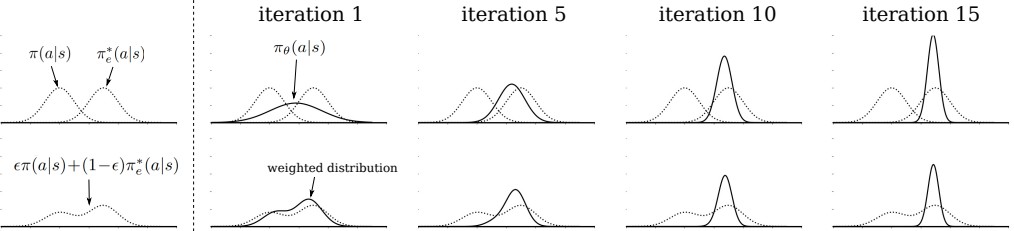

Figure 1: A toy example of the weighted action sampling procedure at each iteration in our algorithm when given a state $s \in S_{e+*}^{\pi_e}$. On both rows, the horizontal lines are the action domains. The left and right dotted lines on the top row describe $\pi \in \Pi \setminus \{\pi_e^*\}$ and $\pi_e^*(a|s)$, respectively. The dotted lines on the bottom row describe the mixture distribution $\pi_e(a|s) = \epsilon\pi(a|s) + (1 - \epsilon)\pi_e^*(a|s)$ with $\epsilon = 0.4$. The solid lines on the top row describe $\pi_\theta(a|s)$ that are optimized with objective (8) at each iteration. The solid lines on the bottom row describe distributions which draw actions, that were already drawn by $\pi_e(a|s)$ in the noisy demonstrations, according to the current importance weight $\pi_\theta(a|s)$ at each iteration. $\pi_\theta(a|s)$ are optimized at each iteration so that the weighted distribution at the previous iteration is the target distribution.

## 5.3 REWARD MAXIMIZATION

As the Off-PAC objective, the objective (9) maximizes the expected (one-step) reward $\hat{R}(s, a) = \pi_{\theta_{old}}(a|s)$. Recall that the learner policy $\pi_\theta(a|s) = \sum_{k=1}^{K} \pi_{\theta^k}(a|s)/K$ is an ensemble of the parameterized policies in our algorithm. Following the work in (Perrone, 1993), we obtain

$$\frac{1}{K} \sum_{k=1}^{K} \mathbb{E}_{s \sim d^{\pi_e}, a \sim \pi_e(\cdot|s)} [\log \pi_{\theta^k}(a|s) \cdot \hat{R}(s, a)] \leq \mathbb{E}_{s \sim d^{\pi_e}, a \sim \pi_e(\cdot|s)} \left[ \log \pi_\theta(a|s) \cdot \hat{R}(s, a) \right], \quad (10)$$

where we use Jensen's inequality with the concave property of logarithm : $\frac{1}{K} \sum_{k=1}^{K} \log \pi_{\theta^k}(a|s) \leq \log \pi_\theta(a|s)$. The inequality (10) indicates that the ensemble of policies $\pi_{\theta^1}, \pi_{\theta^2}, ..., \pi_{\theta^K}$, each of which was learned with (8), has greater or equal values of the objective function in (8) than the averaged values over the policies in the ensemble. As mentioned in 5.2, $\hat{R}(s, a) = \pi_{\theta_{old}}(a|s)$ becomes higher near the mode of $\pi_e^*$. Thus, making $\pi_\theta$ as the ensemble further encourages to shift its mode to that of $\pi_e^*$ and avoid learning $\pi \in \Pi \setminus \{\pi_e^*\}$.

## 5.4 LIMITATIONS

Our algorithm has three limitations. First, $K \times M$ times computational cost is required in comparison with BC, where $M$ is the number of iterations in Algorithm 1. Second, the compounding error due to the probability of non-optimal behavior $\zeta$ still remains unless sufficient amounts of the demonstrations are given. Lastly, $\pi_\theta$ is fitting to $\pi \in \Pi \setminus \{\pi_e^*\}$ rather than $\pi_e^*$ if the major mode of $\epsilon\pi(a|s) + (1 - \epsilon)\pi_e^*(a|s)$ is nearer to the mode of $\pi(a|s)$ than that of $\pi_e^*$. It may be caused due to the higher kurtosis of $\pi(a|s)$ or $\epsilon$ of large values.

## 6 EXPERIMENTS

In our experiments, we aim to answer the following three questions:

Q1. Does our algorithm improve the learner performance more than BC given the noisy demonstrations?

Q2. Can the compounding error due to $\zeta$ be reduced as the number of noisy demonstrations increase?

Q3. Is our algorithm competitive to the existing IL methods if both annotations associated with the non-optimality and environment interactions are allowed?

### 6.1 SETUP

To answer Q1 and Q2, we evaluated our algorithm against BC on four continuous control tasks that are simulated with MuJoCo physics simulator (Todorov et al., 2012). We train an agent on each task by proximal policy optimization (PPO) algorithm (Schulman et al., 2017) using the rewards defined in the OpenAI Gym (Brockman et al., 2016). We use the resulting stochastic policy as the true expert policy $\pi_e^*$. We generate the noisy expert demonstrations using $\pi_e^*$ while randomly adopting non-optimal policies $\pi$ with probabilities of the non-optimal behavior $\epsilon$. The non-optimal policies $\pi$ are selected from uniform distributions $a \sim \mathcal{U}(-u, u)$, Gaussian distributions $a \sim \mathcal{N}(a^*, I)$ with $a \sim \pi_e^*(\cdot|s)$, or a deterministic policy $a = 0$, where $u \in \mathbb{R}^{|\mathcal{A}|}$ denotes all-ones vectors and $I \in \mathbb{R}^{|\mathcal{A}| \times |\mathcal{A}|}$ denotes identity matrices. $\epsilon$ are selected from $\{0.0, 0.1, 0.2, 0.3, 0.4, 0.5\}$. The noisy expert takes actions following $\pi_e^*$ if $z \geq \epsilon$ otherwise $\pi$ which is fixed to a selected one through an episode, where $z \sim \mathcal{U}(0, 1)$. Each noisy demonstration with the selected $\epsilon$ consists of $N$ state-action pairs, where $N$ is selected from $\{5000, 10000, 50000, 100000\}$. Then we perform our algorithm as well as BC to train the learners using each noisy demonstration. We also conducted the same experiments on four low-dimensional discrete control tasks (see Appendix A.4).

To answer Q3, we evaluated our algorithm against IC-GAIL (Wu et al., 2019), 2IWIL (Wu et al., 2019), T-REX (Brown et al., 2019), GAIL and DRIL on three continuous control tasks. IC-GAIL, 2IWIL and T-REX require both annotations associated with the non-optimality and environment interactions. GAIL and DRIL require the environment interactions for the training, but they do not address the noisy demonstration problem. The true expert policy $\pi_e^*$ are obtained in the same way as mentioned above. The non-optimal policies $\pi$ are fixed to $a \sim \mathcal{U}(-u, u)$. We generate the noisy expert demonstrations which consists of 10000 state-action pairs for each $\epsilon \in \{0.05, 0.1, 0.15, ...., 1.0\}$. Then we perform our algorithm and the baselines using all noisy demonstrations. The detailed description of this experimental setup can be found in Appendix A.3.

In both experiments, the performance of the learners is measured by cumulative rewards they earned in an episode. The cumulative reward is normalized with ones earned by $\pi_e^*$ and a random policy $a \sim \mathcal{U}(-u, u)$ so that 1.0 and 0.0 indicate the performance of $\pi_e^*$ and the random policy, respectively. We run five experiments on each task and setup, and measure the mean and standard deviation of the normalized cumulative rewards for each learner over the five experiments. In all experiments, we set the number of policies $K = 5$ in the ensemble learner policy $\pi_\theta$ and the number of iterations $M = 5$. The implementation details of our algorithm can be found in Appendix A.5.

### 6.2 RESULTS

Figure 2 depicts the experimental results against BC. Over all tasks, our algorithm obtains much better learner performance than BC-Single, which is a single (thus not an ensemble) policy learned by BC. It suggests that the policies learned by our algorithm are closer to $\pi_e^*$ than ones learned by BC. The compounding error due to $\zeta$ is expected to be reduced as the number of demonstrations increase. Whereas BC-Ensemble which denotes the ensemble of policies learned by BC yields significant performance gains against BC-Single, increasing the number of noisy demonstrations has a little effect to boost the learner performance trained by BC-Ensemble as shown in Figure 2-(D). It indicates that BC-Ensemble can not reduce the compounding error due to $\epsilon$. On the other hand, our algorithm can boost the learner performance up to that of $\pi_e^*$ as increasing the number of demonstrations. It suggests that our algorithm can reduce the compounding error due to both $\epsilon$ and $\zeta$ if sufficient amounts of the noisy expert demonstrations are given, as is the case for BC with the clean expert demonstrations. The results with the deterministic non-optimal policy $\pi \in \Pi \setminus \{\pi_e^*\}$ which always takes an action $a = 0$ are worse than those with other non-optimal policies. It corresponds to the limitation of our algorithm as mentioned in 5.4, since the major mode of $\epsilon\pi(a|s) + (1-\epsilon)\pi_e^*(a|s)$ might be around $a = 0$. We also conducted ablation experiments where the number of policies $K$ are selected from $\{1, 5\}$ in our algorithm. See Appendix A.6 for details. The ablation experimental results show that the learner obtains better performance if $K$ increases. In addition, the performance of the learner trained by our algorithm is significantly better than that of BC-Single even though $K = 1$. It suggests that our algorithm improves the learner performance by not only the ensemble approach but also using the old policies $\pi_{\theta_{old}}$.

Table 1 shows the experimental results against IC-GAIL, 2IWIL, T-REX, GAIL and DRIL. Over all tasks, 2IWIL and our algorithm can successfully obtain the true expert performance while others can

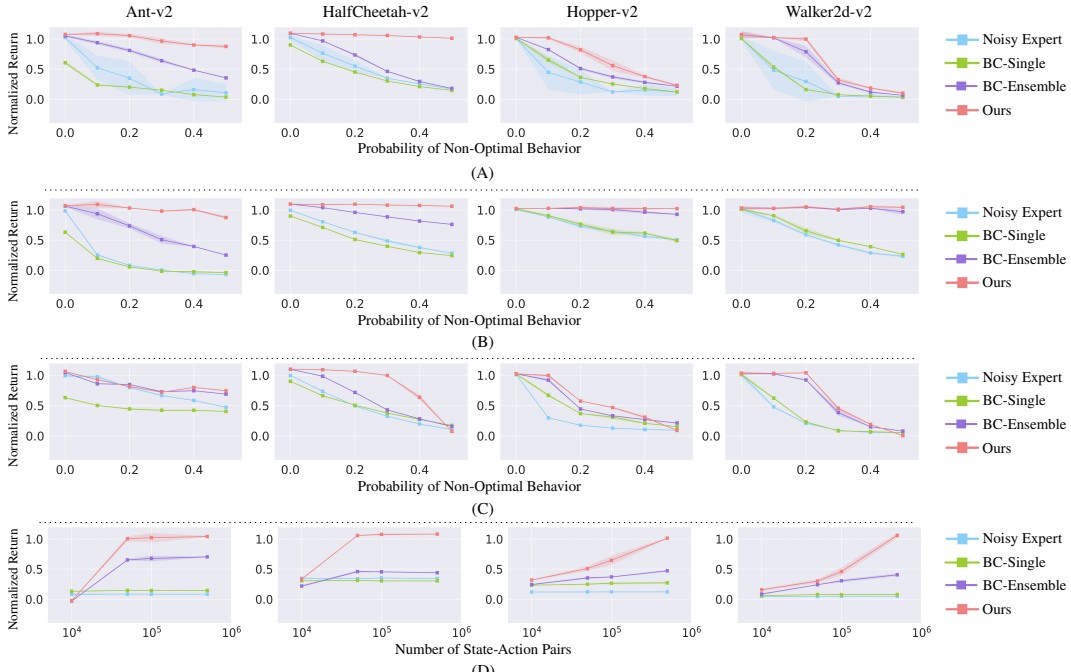

Figure 2: (A)-(C) The performance of policies vs. $\epsilon$ given 50000 state-action pairs of the noisy expert demonstrations where the non-optimal policies $\pi \in \Pi \setminus \{\pi_e^*\}$ are (A) $\mathcal{U}(-u, u)$, (B) $\mathcal{N}(a^*, I)$ with $a \sim \pi_e^*(\cdot|s)$, and (C) the deterministic one $a = 0$, respectively. (D) The performance of policies vs. the number of state-action pairs $N$ of the noisy demonstrations with $\epsilon = 0.3$ where $\pi(a|s) = \mathcal{U}(-u, u)$. BC-Single is a policy learned by BC. BC-Ensemble is an ensemble of policies, each of which was learned by BC. Shaded regions indicate the standard deviation over five experiments.

not. It suggests that our algorithm can obtain competitive results with that of existing IL methods even though the annotation and the environment interactions are not used.

Table 1: The experimental results against IL methods that require the environment interactions.

|  | IC-GAIL | 2IWIL | T-REX | GAIL | DRIL | Ours |
|---|---|---|---|---|---|---|
| Ant-v2 | $0.631 \pm 0.162$ | $1.042 \pm 0.021$ | $0.586 \pm 0.124$ | $0.003 \pm 0.004$ | $1.071 \pm 0.023$ | $1.055 \pm 0.053$ |
| HalfCheetah-v2 | $0.941 \pm 0.103$ | $1.024 \pm 0.059$ | $0.001 \pm 0.113$ | $0.106 \pm 0.003$ | $0.065 \pm 0.006$ | $1.093 \pm 0.092$ |
| Hopper-v2 | $1.233 \pm 0.152$ | $1.223 \pm 0.135$ | $0.441 \pm 0.219$ | $0.000 \pm 0.001$ | $0.910 \pm 0.099$ | $1.003 \pm 0.045$ |

## 7 CONCLUSION

In this paper, we proposed an imitation learning algorithm to cope with the noisy expert demonstrations. Experimental results showed that our algorithm can learn behavior policies that are much closer to the true expert policies than ones learned by BC. Since our algorithm cope well with the noisy expert demonstrations while not requiring any environment interactions and annotations associated with the non-optimal demonstrations, our algorithm is more applicable to real-world problems than the prior works. Although our algorithm has a few limitations as mentioned in 5.4, we believe that the analysis of performance deterioration detailed in Section 4 contributes to step forward for solving the noisy demonstration problems. In future work, we will consider the setting where the probability of non-optimal behavior is state-dependent, which often occurs in the real world more than the state-independent case that we have considered in this paper.

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

# A APPENDIX

## A.1 DETAILED DEVIATION OF THEOREM 1

*Proof.* Let $q_t = (1 - \epsilon)^t$ denotes the probability the noisy expert consecutively follows $\pi_e^*$ in the first $t$ step, and $\chi = \sum_{t=1}^{T} q_{t-1}$ denotes sum of $q_{t-1}$ over time steps. Then we obtain:

$$\mathcal{J}(\pi_e, R) \geq \sum_{t=1}^{T} q_{t-1} R_t^{\pi_e} + (1 - q_{t-1}) \cdot 0 \tag{11}$$

$$\geq T \left\{ \frac{1}{T} \sum_{t=1}^{T} q_{t-1} \right\} \left\{ \frac{1}{T} \sum_{t=1}^{T} R_t^{\pi_e} \right\} \tag{12}$$

$$= \frac{\chi}{T} \left\{ \sum_{t=1}^{T} \mathbb{E}_{s \sim d_t^{\pi_e}} \left[ \epsilon \mathbb{E}_{\pi \sim p_\Pi}[R^\pi(s)] + (1 - \epsilon) R^{\pi_e^*}(s) \right] \right\}$$

$$= \frac{\chi}{T} \left\{ \epsilon \mathbb{E}_{\pi \sim p_\Pi}[\mathcal{J}_{\pi_e}(\pi, R)] + (1 - \epsilon) \mathcal{J}_{\pi_e}(\pi_e^*, R) \right\}$$

$$\geq \frac{\chi}{T} \left\{ \epsilon \mathbb{E}_{\pi \sim p_\Pi}[\mathcal{J}_{\pi_e}(\pi, R)] + (1 - \epsilon) \mathbb{E}_{\pi \sim p_\Pi}[\mathcal{J}_{\pi_e}(\pi, R)] \right\} \tag{13}$$

$$= \left\{ \frac{1}{T} \sum_{t=0}^{T-1} (1 - \epsilon)^t \right\} \cdot \mathbb{E}_{\pi \sim p_\Pi}[\mathcal{J}_{\pi_e}(\pi, R)]$$

The first inequality (11) is from Assumption 2 and 3. The second inequality (12) is from Chebyshev's sum inequality with the monotonically decreasing properties according to Assumption 4. The third inequality (13) is from Assumption 1 : $\mathcal{J}_\beta(\pi, R) \leq \mathcal{J}_\beta(\pi_e^*, R)$ for any $\pi, \beta \in \Pi \setminus \{\pi_e^*\}$. □

## A.2 DETAILED DERIVATION OF THE KL DIVERGENCES

From the definition of (4), we obtain:

$$\mathbb{E}_{s \sim d^{\pi_e}, a \sim \pi_e(\cdot|s)}[\log \pi_\theta(a|s)] = \alpha \mathbb{E}_{s \sim d_e^{\pi_e}, a \sim \pi_e(\cdot|s)}[\log \pi_\theta(a|s)] \tag{14}$$

$$+ \beta \mathbb{E}_{s \sim d_{e+*}^{\pi_e}, a \sim \pi_e(\cdot|s)}[\log \pi_\theta(a|s)] \tag{15}$$

The forward Kullback-Leibler (KL) divergence $D_{KL}$ between $\pi_e$ and $\pi_\theta$ over a state distribution $d^{\pi_e}$ is defined as $\mathbb{E}_{s \sim d^{\pi_e}}[D_{KL}(\pi_e(\cdot|s)||\pi_\theta(\cdot|s))] = -\mathbb{E}_{s \sim d^{\pi_e}}[\mathbb{E}_{a \sim \pi_e(\cdot|s)}[\log \pi_\theta(a|s)] + H[\pi_e(\cdot|s)]]$, where $H$ denotes the entropy. Since $H[\pi_e(\cdot|s)]$ always takes positive value and is not associated with $\theta$, we obtain an inequality : $\mathbb{E}_{s \sim d^{\pi_e}, a \sim \pi_e(\cdot|s)}[\log \pi_\theta(a|s)] \leq -\mathbb{E}_{s \sim d^{\pi_e}}[D_{KL}(\pi_e(\cdot|s)||\pi_\theta(\cdot|s))]$. The same goes with (14) as

$$\alpha \mathbb{E}_{s \sim d_e^{\pi_e}, a \sim \pi_e(\cdot|s)}[\log \pi_\theta(a|s)] \leq -\alpha \mathbb{E}_{s \sim d^{\pi_e}}[D_{KL}(\pi_e(\cdot|s)||\pi_\theta(\cdot|s))]. \tag{16}$$

Since $\pi_e$ adopts both $\pi_e^*$ and $\pi \in \Pi \setminus \{\pi_e^*\}$ following the probability $\epsilon$, the third term (15) can be expanded as:

$$\beta \mathbb{E}_{s \sim d_{e+*}^{\pi_e}, a \sim \pi_e(\cdot|s)}[\log \pi_\theta(a|s)] = \beta \mathbb{E}_{s \sim d_{e+*}^{\pi_e}} \left\{ \epsilon \mathbb{E}_{\pi \sim p_\Pi, a \sim \pi(\cdot|s)}[\log \pi_\theta(a|s)] \right.$$

$$\left. + (1 - \epsilon) \mathbb{E}_{a \sim \pi_e^*(\cdot|s)}[\log \pi_\theta(a|s)] \right\}$$

$$\leq -\beta \left\{ \epsilon \mathbb{E}_{s \sim d_{e+*}^{\pi_e}, \pi \sim p_\Pi}[D_{KL}(\pi(\cdot|s)||\pi_\theta(\cdot|s))] \right.$$

$$\left. + (1 - \epsilon) \mathbb{E}_{s \sim d_{e+*}^{\pi_e}}[D_{KL}(\pi_e^*(\cdot|s)||\pi_\theta(\cdot|s))] \right\} \tag{17}$$

## A.3 DETAILED DESCRIPTION OF THE EXPERIMENTAL SETUP

We annotate confidence scores for the noisy demonstrations so that the confidence is one if the demonstrations are obtained with $\epsilon = 0$ otherwise zero. The confidence scores are used IC-GAIL as well as 2IWIL. We use publicly available code [2] for the implementation of both IC-GAIL and

---

[2] https://github.com/kristery/Imitation-Learning-from-Imperfect-Demonstration

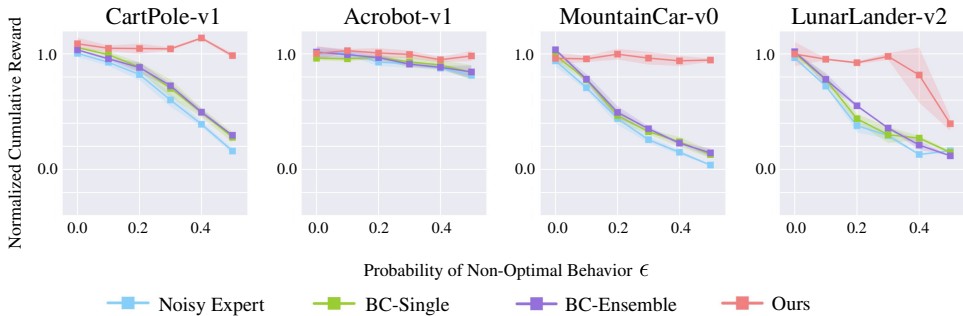

Figure 3: The performance of policies vs. $\epsilon$ given 50000 state-action pairs of the noisy expert demonstrations where the non-optimal policies $\pi \in \Pi \setminus \{\pi_e^*\}$ select actions uniformly at random. BC-Single is a policy learned by BC. BC-Ensemble is an ensemble of policies, each of which was learned by BC. Shaded regions indicate the standard deviation over five experiments.

2IWIL. We follow the training procedure of both methods as described in Section 5 in (Wu et al., 2019).

We annotate rankings for the noisy demonstrations so that the smaller $\epsilon$ correspond to higher rankings. Then, we train the learner by T-REX given the ranked demonstration data. We use publicly available code [3] for the implementation of T-REX.

For training the learner with GAIL and DRIL, we use all noisy demonstrations without any screening process. We use publicly available code [4] for the implementation of GAIL and DRIL.

### A.4 EXPERIMENTAL RESULTS ON DISCRETE CONTROL TASKS

Figure 3 shows the experimental results on four discrete control tasks. Over all tasks, our algorithm obtain much better results than BC.

### A.5 IMPLEMENTATION DETAILS OF OUR ALGORITHM

We implement our algorithm using $K$ neural networks with two hidden layers to represent policies $\pi_{\theta^1}, \pi_{\theta^2}, ..., \pi_{\theta^K}$ in the ensemble. The input of the networks is vector representations of the state. Each neural network has 100 hidden units in each hidden layer followed by hyperbolic tangent nonlinearity, and the dimensionality of its final output corresponds to that of action space. The final output is followed by softmax function in the discrete control tasks. As for the continuous control tasks, the final output represents the mean of a Gaussian policy as $\pi_{\theta^k} = \mathcal{N}(\mu_{\theta^k}(s), \sigma_{\theta^k}^2)$, where $\sigma_{\theta^k}^2$ is implemented as a trainable independent vector from the networks. The neural network architecture for the policy trained by BC is the same as the ones for a single policy in our algorithm. We employ Adam (Kingma & Ba, 2014) for learning parameters with a learning rate of $\eta * 10^{-4}$ where $\eta = K / \sum_{k=1}^{K} \pi_{\theta_{old}^k}(\mu_{\theta_{old}^k}(s)|s)$ is a scaling parameter. The parameter $\eta$ plays a role in scaling $\hat{R} = \pi_{\theta_{old}}(a|s)$ to avoid the training being slow due to $\pi_{\theta_{old}}(a|s)$ of small values.

The parameters in all layers are initialized by Xavier initialization (Glorot & Bengio, 2010). The mini-batch size and the number of training epochs are 128 and 500, respectively.

### A.6 ABLATION EXPERIMENTS

We conducted ablation experiments where we evaluate how the number of policies $K$ in the ensemble policy $\pi_\theta$ as well as the number of the policies $K_{old}$ used in the *old* ensemble policies $\pi_{\theta_{old}}$ affect the performance. Table 2 summarizes the ablation experimental results. Even if our algorithm uses $K = 1$ as BC-Single does, the results of our algorithm are better than BC. It indicates that the

---

[3] https://github.com/hiwonjoon/ICML2019-TREX
[4] https://github.com/xkianteb/dril

weighted action sampling described in 5.2 works to avoid learning the non-optimal policies without relying on the ensemble approach. The same goes with $K = 5$. Our algorithm with $K = 5$ and $K_{old} = 1$ obtain much better performance than BC-Ensemble with $K = 5$. This result also supports the weighted action sampling works. The learner performance with fixed $K$ increases as $K_{old}$ increases. Similarly, the learner performance with fixed $K_{old}$ increases as $K$ increases. It suggests that both $K$ and $K_{old}$ affect the performance in our algorithm.

Table 2: The performance of policies on the ablation experiment. The number of state-action pairs of the noisy expert demonstrations is $N = 50000$. The non-optimal policies $\pi \in \Pi \setminus \{\pi_e^*\}$ is $\mathcal{U}(0, I)$. BC-Single is a policy learned by BC. BC-Ensemble is an ensemble of five policies, each of which was learned by BC. $K$ denotes the number of policies in the ensemble policy $\pi_\theta$. $K_{old}$ denotes the number of policies used in the old ensemble policy $\pi_{\theta_{old}}$. The mean and standard deviation of the normalized cumulative rewards over three experiments are described.

| | Ant-v2 | HalfCheetah-v2 | Hopper-v2 | Walker2d-v2 | Average |
|---|---|---|---|---|---|
| BC-Single | $0.149 \pm 0.001$ | $0.305 \pm 0.006$ | $0.258 \pm 0.017$ | $0.071 \pm 0.004$ | $0.196 \pm 0.105$ |
| BC-Ensemble($K = 5$) | $0.664 \pm 0.043$ | $0.459 \pm 0.014$ | $0.352 \pm 0.028$ | $0.279 \pm 0.039$ | $0.438 \pm 0.167$ |
| Ours($K = 1, K_{old} = 1$) | $0.272 \pm 0.184$ | $0.505 \pm 0.279$ | $0.405 \pm 0.206$ | $0.281 \pm 0.306$ | $0.366 \pm 0.111$ |
| Ours($K = 5, K_{old} = 1$) | $0.903 \pm 0.048$ | $1.057 \pm 0.008$ | $0.602 \pm 0.130$ | $0.345 \pm 0.049$ | $0.731 \pm 0.320$ |
| Ours($K = 1, K_{old} = 5$) | $0.517 \pm 0.015$ | $0.907 \pm 0.007$ | $0.778 \pm 0.093$ | $0.414 \pm 0.085$ | $0.654 \pm 0.227$ |
| Ours($K = 5, K_{old} = 5$) | $0.995 \pm 0.053$ | $1.058 \pm 0.053$ | $0.573 \pm 0.079$ | $0.364 \pm 0.044$ | $0.747 \pm 0.334$ |

