# OpenReview forum: "Behavioral Cloning from Noisy Demonstrations"
_ICLR.cc/2021/Conference — ICLR 2021 Spotlight_

### Official Review · AnonReviewer2 · 2020-10-28

**Rating:** 6
**Confidence:** 4

**Review:**

---
Summary

This paper studies the problem of imitation an expert from noisy demonstrations without interactions with the environment. It proposes an algorithm, which utilizes an ensemble of behavioral cloning policies and is analogous to the mean shift algorithm, to find the mode from noisy demonstrations. Experimentally, the proposed algorithm can almost recover the expert policy.

---
Writing Quality

The writing quality of the paper can be further improved, especially in Section 4.3. I'd appreciate if the authors can indicate what policy is used in Figure 1.

Please make the notations consistent. Are $S_{\pi_e}$ and $S^{\pi_e}$ the same?

---
Comments

Theorem 1 seems to be incorrect. Assuming $T$ is large enough and $\epsilon$ is small enough so that $\sum_{i=0}^{T-1}(1 - \epsilon)^i > 1$, and $p_\Pi$ is a Dirac distribution at $\pi_e$, then Theorem 1 says $J(\pi_e, R) > J_{\pi_e}(\pi_e, R)$. I think the reason is that a factor of $T$ is missing from the 3rd line to the 4th line in the proof. In addition, the theorem only provides a lower bound of performance, so the sentence "It also shows that BC can not boost ... Assumption 1" in Section 4.1 requires more justification.

I don't understand the argument in Section 4.3. What's the difference between $S^{\pi_e}_{e}$ and $S^{\pi_e}_{e+*}$?

I don't see the meaning of divergence (5) and (6), either. The objective (1) can be just expressed by something similar to divergence (7), like
$$
(1) = \epsilon E_{s \sim d^{\pi_e}, \pi \sim p_\Pi}[KL(\pi(\cdot|s)|\pi_\theta(\cdot | s))] + (1 - \epsilon) E_{s \sim d^{\pi_e}}[KL(\pi^*_e(\cdot|s)|\pi_\theta(\cdot|s))].
$$

> Since the objective of IL is to obtain policies that can generate or keep being on optimal state trajectories as the true expert does, minimizing the divergence (5) over the states that are out of the optimal trajectories have a little effect to learn the $\pi_e^*$.

By this argument, if we remove $S^{\pi_e}_e$ from the training set, will the performance change much? Also, the experiments uses $\epsilon$ as large as 0.4, so most states are in $S^{\pi_e}_e$, if I didn't misunderstand it.

As the authors also note later, the proposed algorithm doesn't resolve the compounding error issue. Minimizing divergence (5) can somehow make a correction. I think the reason why more data can help is that more data covers more states, so the correction effect is stronger.

Other comments:
1. Assumption 4 seems a little bit weird. What does it make sense?
2. What does U(0, I) mean?
3. What's the setting in Table 1? Why the number of Hopper-v2 is quite different from Table 2?
4. L6 of the algorithm: is the objective (1) or (8)?
5. It seems that objective (9) is linear to $\eta$, so I don't understand how $\eta$ works.

Also I'd like to see more discussions in the related work, e.g.,
1. Conservative Q-Learning for Offline Reinforcement Learning, by Kumar et al.
2. Learning Self-Correctable Policies and Value Functions from Demonstrations with Negative Sampling, by Luo et al.
3. Off-Policy Deep Reinforcement Learning without Exploration, by Fujimoto et al.
4. Deep Q-learning from Demonstrations, by Hester et al.

---
UPDATED After Rebuttal

Thanks the authors for answering my questions.

Theorem 1 now looks fine.

**It can be said that BC is to maximizes the off-policy expected...**

I do understand that the existence of $\epsilon$ makes the RHS of Eq (3) (All labels are in the revised version. ) smaller. However, Eq (3) only means the maximum *guarantee* (the lowest performance) of BC is small, and doesn't mean that the BC's performance can't exceed something.

**definitions of $S_e^{\pi_e}$ and $S_{e+*}^{\pi_e}$**

Thanks for clarification. It's much clearer now. However, in this definition, how about the states where the policy has always followed the expert policy so far? I guess this is the reason why you have 3 sets before revision.

**The first term on the RHS in (7) thus works as a “noisy” regularizer.** (in the revised paper)

In my opinion, it goes a little bit beyond regularization, as the coefficient is as large as 0.4. For example, in the image classification task, can we call corrupting 40% labels a "noisy regularization"?

**It could happen that removing $S_e^{\pi_e}$ from the training set changes the performance in such a case where the noisy expert recovers to be in the optimal state after adopting the non-optimal policies.**

But with much larger probability, it will go out of the manifold (of expert policy's states), right? Even BC can go out of it easily (otherwise you can't beat it).

Overall, I think this is a neat algorithm and it seems to work pretty well. The revision also makes it much easier to understand and I do appreciate the effort behind it. I'll then raise my current score (5) to 6, although I'm still not fully convinced so won't give a higher score.

---

> ### Author Response · Authors · 2020-11-20
> **Response to AnonReviewer2**
>
> We thank the reviewer for the detailed and constructive feedback. Following comments from all reviewers, we revised our manuscript and submitted it. It would be appreciated that if the reviewer could read the revised manuscript.
>
> **Theorem 1 seems to be incorrect**.
>
> As the reviewer pointed out, we had omitted factor $\frac{1}{T}$ in Theorem 1, Corollary 1, and the proof. We have fixed the issue in the revised manuscript.
>
> **In addition, the theorem only provides a lower bound of performance, so the sentence "It also shows that BC can not boost ... Assumption 1" in Section 4.1 requires more justification**.
>
> It can be said that BC is to maximizes the off-policy expected $T$-step reward $J_{\pi_e}(\pi_{\theta},R)$. The Equation (2) (Equation (3) in the revised manuscript) shows that BC is to maximizes the on-policy expected $T$-step reward $J(\pi_{\theta},R)$ through maximizing $J_{\pi_e}(\pi_{\theta},R)$. The Equation (2) the maximization is significantly negated as $\epsilon$ increases. Hence, the learner trained by BC is not able to boost the learner performance enough if the noisy demonstrations were given.
>
> We have added the above explanation in subsections 4.1 and 4.2.
>
> **I don't understand the argument in Section 4.3. What's the difference between $S_e^{\pi_e}$ and $S_{e+\ast}^{\pi_e}$ ?**
>
> As all reviewer have pointed out, the description in subsection 4.3 was misleading. We revised the overall descriptions in subsection 4.3 to clarify what we would like to claim through the analysis in terms of KL divergence minimization. Please see the revised manuscript. We have rewritten the explanation of $S_e^{\pi_e}$ and $S_{e+\ast}^{\pi_e}$ as follows.
>
> *$S^{\pi_e}$ can be defined with two state sets of states as $S^{\pi_e}={S_e^{\pi_e}}\cup{S_{e+\ast}^{\pi_e}}$, where ${S_e^{\pi_e}}$ contains states that are observed if the noisy expert has followed a policy $\pi\in\Pi\setminus\{\pi^\ast_e\}$ once or more so far in the episode, and ${S_{e+\ast}^{\pi_e}}$ contains states at which the noisy expert has followed a policy $\pi$ at the first time in the episode.*
>
> **I don't see the meaning of divergence (5) and (6), either**.
>
> We think that considering the KL divergence on each subset enable to analyze the performance deterioration in more details. For example, the following argument can be made as described in the revised manuscript.
>
> *As mentioned in 4.1 and 4.2, BC is to maximize $J_{\pi_e}(\pi_{\theta},R)$ to $J(\pi_e,R)$. Hence, minimizing $\Omega_e(\theta)$ in (6) corresponds to maximize $E_{s\sim d_e^{\pi_e}}[R^{\pi_\theta}(s)]$ to $E_{s\sim d_e^{\pi_e}}[R^{\pi_e}(s)]$. Since the rewards $R^{\pi_e}(s)$ are at least zero for the states $s\sim d_e^{\pi_e}$ according to Assumption 3 and the definition of $S_e^{\pi_e}$, $E_{s\sim d_e^{\pi_e}}[R^{\pi_\theta}(s)]$ becomes at least zero by minimizing $\Omega_e(\theta)$.*
>
> **By this argument, if we remove $S_e^{\pi_e}$ from the training set, will the performance change much?**
>
> As far as we could say, the rewards $R^{\pi_e}(s)$ are at least zero for the states $s\in S_e^{\pi_e}$. It could happen that removing $S_e^{\pi_e}$ from the training set changes the performance in such a case where the noisy expert recovers to be in the optimal state after adopting the non-optimal policies.
>
> **1.Assumption 4 seems a little bit weird. What does it make sense?**
>
> Since the probability the noisy expert consecutively follows $\pi_e^\ast$ decreases as time step $t$  increases according to Assumption 2, the divergence between $d_t^{\pi_e}$ and $d_t^{\pi_e^\ast}$ becomes greater as $t$  increases, and thus the one-step "expected" immediate reward $R^{\pi_e}_t$ decreases as $t$ increases.
>
> We have added the above explanation in Section 3.
>
> **2. What does U(0, I) mean?**
>
> We have clarified the notation in Section 6.
>
> **3. What's the setting in Table 1? Why the number of Hopper-v2 is quite different from Table 2?**
>
> The setting in Table 1 can be found in 6.1. Since the number of the noisy demonstrations are different between the experiments, the number of Hopper-v2 is different between Table 1 and Table 2.
>
> **4. L6 of the algorithm: is the objective (1) or (8)?**
>
> L6 of the algorithm should be the Equation (9). We have fixed it in the revised manuscript.
>
> **5. It seems that objective (9) is linear to $\eta$, so I don't understand how works.**
>
> As the dimension of the action space increases, $\pi_{\theta_{old}}(a|s)$ for each action decreases. The parameter $\eta$ plays a role in scaling $\hat{R}=\pi_{\theta_{old}}(a|s)$ to avoid the training being slow due to $\pi_{\theta_{old}}(a|s)$ of small values.
> We have added the explanation above in Appendix 5.
>
> **Also I'd like to see more discussions in the related work,**
>
> We have added discussions on offline RL methods (e.g. Conservative Q-Learning for Offline Reinforcement Learning) in Section 2.
>
> Again, thank you for your valuable comments. Please do let us know if you have any remaining questions/comments.
>
> Best, Authors

---

### Official Review · AnonReviewer1 · 2020-10-29

**Rating:** 7
**Confidence:** 3

**Review:**

Summary
--
This paper proposes an IL method for learning from noisy demonstrations. The paper develops a bit of theory to illustrate an upper-bound on BC performance in the noisy observation setting, and uses this bound to motivate the construction of an algorithm.

Experiments illustrate that the method achieves superior performance to standard BC and comparable performance to some prior IL works that require interactive environment access, among other results.

Clarity and Correctness
--
There are two major specific clarity issues, along with general ambiguity due to a significant amount of grammatical errors and some mathematical errors.

(1) There are issues with the theorem and its proof. First, a minor point, "Corollary 1" has no accompanying theorem, rename this to "Theorem 1" (or is it actually supposed to be a Corollary of Theorem 1, which is actually in the appendix? I would expect the Theorem, not the Corollary, to be in the main text). Eq (2) ("corollary 1") and Eq (11) are different (NB \hat \epsilon and the expectation and the LHS!), thus the proof is proving something else besides Eq (2) holding. Further, Eq (12) is not clearly connected to any definitipon in the paper. The explanation indicates that Assumptions 2&3 are combined, along with the definition, in a single step, but is far from clear and therefore far from being clearly correct.

(2) Section 5 does not clearly connect the algorithm with the results in Section 4. This may be due to the fact that Section 5 seems to be presented backwards, with the main result first, and places the reasoning that connects Section 4 to Section 5 at the end of Section 5, rather than the beginning.

Other clarity issues
- In S4.2, it's not clear what the constraints on \zeta are, or how it's connected to Eq (1).
- If my understanding is correct, it would be clearer to explicitly say that BC from obs is difficult because the maximization of the LHS of (4), i.e. the BC objective, is upper-bounded by the difficult-to-optimize RHS of (4).
- Alg 1 L6 -- should this be objective (8) or (9)? [not 1, because (1) doesn't include \hat R] Also, \bar\pi is undefined in the Alg 1.
- I cannot interpret the bottom row of Fig 2 becuase the caption is ambiguous.
- I cannot parse the last line of 5.4, this requires a clearer explanation.

Originality
--
- One closely related work that is undiscussed is Disagreement-Regularized Imitation Learning (DRIL) (Bratley et al., ICLR 2020), which also incorporates policies trained with BC into a new training objective to regularize the training of a new policy. Quantitative and qualitative comparison to DRIL would give the paper more context and clearer originality.

Significance
--
The paper presents a method that demonstrates compelling performance in a few different settings, particularly in that it is competitive with IL methods that require environment interaction.

Other comments
--
- abstract: "but the non-optimal" -> "but not non-optimal" ?
- I cannot unerstand the sentence at the top of p3: "We refer to J (π, R) as on-policy expected T-step reward since the policy the expected reward is evaluated for and the one inducing the state distribution dπ are the same"
- Replace the comma in Assumption 1 with a semicolon or other separator, currently it could be misinterpreted to mean J(\pi, R) \leq J_\beta(\pi, R).

---

> ### Author Response · Authors · 2020-11-20
> **Response to AnonReviewer1**
>
> We thank the reviewer for the detailed and constructive feedback. Following comments from all reviewers, we revised our manuscript and submitted it. It would be appreciated that if the reviewer could read the revised manuscript.
>
> **There are two major specific clarity issues, along with general ambiguity due to a significant amount of grammatical errors and some mathematical errors**.
>
> We have fixed those errors in the revised manuscripts.
>
> **(1) There are issues with the theorem and its proof.**
> **First, a minor point, "Corollary 1" has no accompanying theorem, rename this to "Theorem 1"**.
>
> We have put the description of Theorem 1 in the main text.
>
> **Eq (2) ("corollary 1") and Eq (11) are different(NB $\hat{\epsilon}$ and the expectation and the LHS!), thus the proof is proving something else besides Eq (2) holding.**
>
> We have replaced $C$ for $R$ on the LHS in Equation (12) (Equation (11) in the revised manuscript) so that Theorem 1 (Equation (2) in the revised manuscript) and Equation (12) are the same. The derivation of Corollary 1 from Theorem 1 can be found in Appendix A.1 of the manuscript before revision and in subsection 4.2 of the revised manuscript.
>
> **Further, Eq (12) is not clearly connected to any definitipon in the paper**.
>
> It can be said that BC is to maximizes the off-policy expected $T$-step reward $J_{\pi_e}(\pi_{\theta},R)$. The Equation (2) (Equation (3) in the revised manuscript) shows that BC is to maximizes the on-policy expected $T$-step reward $J(\pi_{\theta},R)$ through maximizing $J_{\pi_e}(\pi_{\theta},R)$. The Equation (2) the maximization is significantly negated as $\epsilon$ increases. Hence, the learner trained by BC is not able to boost the learner performance enough if the noisy demonstrations were given.
>
> We have added the above explanation in subsections 4.1 and 4.2.
>
> **(2) Section 5 does not clearly connect the algorithm with the results in Section 4**.
>
> In the beginning of Section 5, we have added descriptions why the learner trained by BC is not able to boost its performance and how our algorithm resolves the issues.
>
> **In S4.2, it's not clear what the constraints on $\zeta$ are, or how it's connected to Eq (1)**.
>
> If the optimization of BC was difficult, the learner may adopt the non-optimal policies with probability more than $\epsilon$. $\zeta$ is the additional probability of the non-optimal behavior due to the difficulty in the optimization. Then the learner with the probability of the non-optimal behavior $\hat{\epsilon}=\epsilon+\zeta$ can be deemed as another noisy expert. Thus Theorem 1 can be applied to the learner as the noisy expert and Corollary 1 for the learner can be derived from Theorem 1.
>
> **If my understanding is correct, it would be clearer to explicitly say that BC from obs is difficult because the maximization of the LHS of (4), i.e. the BC objective, is upper-bounded by the difficult-to-optimize RHS of (4)**.
>
> We totally agree with it. We have added descriptions as the reviewer suggested in subsection 4.3.
>
> **Alg 1 L6 -- should this be objective (8) or (9)?**
>
> Alg 1 L6 should be the Equation (9). We have fixed it in the revised manuscript.
>
> **I cannot interpret the bottom row of Fig 2 because the caption is ambiguous**.
>
> We have added explanations in the caption to avoid the ambiguity as follows:
>
> *The solid lines on the top row describe $\pi_\theta(a|s)$ that are optimized with objective (8) at each iteration. The solid lines on the bottom row describe distributions which draw actions, that were already drawn by $\pi_e(a|s)$ in the noisy demonstrations, according to the current importance weight ${\pi}_{\theta}(a|s)$ at each iteration. $\pi_\theta(a|s)$ are optimized at each iteration so that the weighted distribution at the previous iteration is the target distribution*.
>
> **I cannot parse the last line of 5.4, this requires a clearer explanation.**
>
> If the major mode of $\epsilon\pi(a|s)+(1-\epsilon)\pi_e^\ast(a|s)$ is nearer to the mode of $\pi(a|s)$ than that of $\pi_e^\ast$.
> It may be caused due to the higher kurtosis of $\pi(a|s)$ or $\epsilon$ of large values.
> We have added the above explanations.
>
> **Quantitative and qualitative comparison to DRIL would give the paper more context and clearer originality**.
>
> We have added the quantitative and qualitative comparison to DRIL in Section 6 and 2, respectively. The experimental results in Section 6 show that superiority of our algorithm to DRIL.
>
> **Other comments**
>
> We have fixed the errors that the reviewer pointed out.
>
> Again, thank you for your valuable comments. Please do let us know if you have any remaining questions/comments.
>
> Best, Authors

---

> > ### Comment · AnonReviewer1 · 2020-11-20
> > **Responses**
> >
> > I thank the authors for addressing my concerns and improving the manuscript's clarity. I have read the concerns of the other reviewers, and believe that the authors have addressed them. I will increase my score one point from a 6 to a 7.
> >
> > Some final suggestions -- it would be nice if the Algorithm pseudocode were more detailed in order to make the Algorithm self-contained. (1) Line 6 is vague, instead be precise about how the approximate maximization is taking place. Since the paper is in single-column format, you have plenty of horizontal space to make use of here. (2) The algorithm should return $\pi_{\theta^K}(a|s)$, no? (3) Change the title from "Overview of Our Algorithm" to something informative, e.g. "Behavior Cloning from Noisy Demonstrations".

---

> > > ### Author Response · Authors · 2020-11-24
> > > **Response to AnonReviewer1**
> > >
> > > We really appreciate that the reviewer raised the score and provided additional suggestions for improving our manuscript. Following the suggestions, we have revised our manuscript. It would be appreciated if the reviewer could check the revised manuscript once more again.
> > >
> > > **(1) Line 6 is vague, instead be precise about how the approximate maximization is taking place.**
> > >
> > > We have rewritten the optimization procedure in more detail.
> > >
> > > **(2) The algorithm should return $\pi_{\theta^K}(a|s)$, no?**
> > >
> > > Our algorithm returns the ensemble $\pi_\theta$ where $\pi_{\theta_K}$ is a policy in the ensemble. We have added a line which explicitly explains that our algorithm returns $\pi_\theta$.
> > >
> > > **(3) Change the title from "Overview of Our Algorithm" to something informative**
> > >
> > > We have changed the title of Algorithm 1 to "Behavior Cloning from Noisy Demonstrations".
> > >
> > >
> > > Best,
> > >
> > > Authors

---

### Official Review · AnonReviewer4 · 2020-10-29
**Nice and efficient idea, the analysis needs some clarification.**

**Rating:** 8
**Confidence:** 4

**Review:**

This paper introduces a new offline imitation learning algorithm, to be compared with BC.

The main idea (and difference with simple BC) is to re-use the old learned policy as a reward function in a policy-gradient step.
As an effect, the action distribution is bootstraped in a way that enforces the action near the mode of the observed demonstration.
Consequently, if the demonstrations are noisy but the mode of the optimal behaviour is conserved, then it will learn a policy closer to the optimal behaviour than the demonstration itself.

I think this idea is original, simple and smart. The empirical results show that it works very well on classic locomotion tasks with artificial noises (uniform/gaussian). I only regret two things:

1) The first part (theoretical) of the paper, and especially section 4, was unclear to read and somehow misleading :

- In section 3 assumption 3, if the noisy expert has never followed the noise, then the reward should not depend on the noise. Maybe there is an inversion of the formulation here. And I don’t get at all the link between assumption 3 and its interpretation in the paragraph above.

- In section 4, I don’t understand the utility of equation 2) regarding the rest of the paper. Also, as it is a lower bound of the performance of BC, I don’t get how it can show any limitation of this approach (I would have expected an upper bound in that perspective).

- I am still not sure to understand what is the state partition (despite the trial to explain with a figure). It looks like the expert demonstration must necessary start without noise (generating states belonging to S_*) then the very first time the action is sampled from the noise the state belongs to S_e+*, after which all the states belong to S_e.

2) It would have been interesting to observe experiments with more chaotic noises (for example generated by biased learning algorithms or real-world human demonstrations). But this last point only reflects my frustration, the paper is already showing sufficient results to be impactful.

---

> ### Author Response · Authors · 2020-11-20
> **Response to AnonReviewer4**
>
> We thank the reviewer for the positive and constructive feedback. Following comments from all reviewers, we revised our manuscript and submitted it. It would be appreciated that if the reviewer could read the revised manuscript.
>
> **In section 3 assumption 3, if the noisy expert has never followed the noise, then the reward should not depend on the noise.**
>
> As the reviewer mentioned, the reward should not depend on $\epsilon$ if the noisy expert has never followed the non-optimal policies. On the other hand, the "expected" one-step immediate reward is supposed to be ${R}^{\pi_e}=E_{s\sim d_t^{\pi_e}} \bigl[ \epsilon E_{\pi \sim p_{\Pi}}  [{R}^{\pi}({s})]  +(1-\epsilon) R^{\pi_e^\ast}({s}) \bigr] $ if the noisy demonstrations have a number of episodes where the noisy expert has reached the same state $s$ at time step $t$.
>
> **And I don’t get at all the link between assumption 3 and its interpretation in the paragraph above.**
>
> To make the description interpretable, we have rewritten the description in Section 3 as follows:
>
> *The noisy expert is going to visit states that would be never visited by the true expert if the noisy expert followed non-optimal policies even once. Assumption 3 indicates that those states are less rewarded and their rewards are at least zero.  Assumption 3 also indicates that the noisy demonstrations have a number of episodes where the noisy expert has reached the same state $s$ where the noisy expert has adopted $\pi,\beta\in\Pi\setminus\{\pi^{\ast}_e\}$ with the probability $\epsilon$.*
>
> **In section 4, I don’t understand the utility of equation 2) regarding the rest of the paper. Also, as it is a lower bound of the performance of BC, I don’t get how it can show any limitation of this approach (I would have expected an upper bound in that perspective).**
>
> It can be said that BC is to maximizes the off-policy expected $T$-step reward $J_{\pi_e}(\pi_\theta, R)$. The Equation (2) (Equation (3) in the revised manuscript) shows that BC is to maximizes the on-policy expected $T$-step reward ${ J}(\pi_{\theta},R)$ through maximizing ${J}_{\pi_e}(\pi_{\theta},{ R})$. The equation (2) the maximization is significantly negated as $\epsilon$ increases. Hence, the learner trained by BC is not able to boost the learner performance enough if the noisy demonstrations were given.
>
> We have added the above explanation in subsections 4.1 and 4.2.
>
> **It looks like the expert demonstration must necessary start without noise (generating states belonging to $S_\ast$) then the very first time the action is sampled from the noise the state belongs to $S_{e+\ast}$, after which all the states belong to $S_e$.**
>
> What the reviewer mentioned is just what we would like to mention.
>
> **I am still not sure to understand what is the state partition (despite the trial to explain with a figure)**.
>
> As all reviewer have pointed out, the description in subsection 4.3 was misleading. We revised the overall descriptions in subsection 4.3 to clarify what we would like to claim through the analysis in terms of KL divergence minimization. Please see the revised manuscript.
>
> **It would have been interesting to observe experiments with more chaotic noises (for example generated by biased learning algorithms or real-world human demonstrations)**.
>
> We will conduct experiments with more chaotic noises and add the result in the camera ready version.
>
> Again, thank you for your valuable comments. Please do let us know if you have any remaining questions/comments.
>
> Best, Authors

---

### Decision · Program_Chairs · 2021-01-07
**Final Decision**

**Decision:**

Accept (Spotlight)

**Comment:**

This paper focuses on the problem of performing imitation learning from trajectory-level data that includes optimal as well as suboptimal demonstrations.  The authors wish to avoid the requirement of a separate filtering process that would throw away the bad trajectories.  The authors propose a clever innovation that allows for leveraging the policy that is itself being learned to reweight the samples for a next round of weighted behavioral cloning.  The paper is also somewhat theoretically rigorous and provides insight into the problem.

The reviewers pointed out some initial issues related to clarity and the authors did a good job of addressing reviewer concerns.  Ultimately all reviewers agreed that the core innovation of the paper was interesting and empirically worked reasonably well.

One older line of work that I think is quite relevant, but which is not discussed, is the empirically observed "clean-up effect", described by Michie and colleagues in the 90s (e.g. "Learning to fly" Sammut et al 1992).  This clean-up effect is intuitive and reportedly achieved for free in settings where the learning objective is mode-seeking and the dataset is large, insofar as the mean value of the resulting policy *should* produce actions that corresponds to the average action produced by demonstrators in the same situation.  I think it would be worth discussing how the analysis of this paper relates to this empirical phenomenon. In particular, it would be worth clarifying in what regimes the suboptimality of training from a dataset with noisy examples arises and how likely this is to effect the mean value of the learned policy (for context, it is fairly common in practice to evaluate the student policy in BC settings by only using the mean action value; perhaps this point was present in the paper, and I missed it). From a certain perspective, the innovation of this paper is to accentuate the clean-up effect.

As noted by a reviewer, and subsequently incorporated into the paper, the actual algorithm has some similarities to versions of recent "offline RL" algorithms (though of course it does not leverage rewards).  In particular, the motif of performing a weighted regression could perhaps be a bit more thoroughly contextualized by connecting it to other weighting factors (e.g. see Critic Regularized Regression).  That said, I leave this entirely to the discretion of the authors.

The final scores were 8, 7, & 6.  I see this as a strong paper and will endorse it for a spotlight.